# *Candida* Cell-Surface-Specific Monoclonal Antibodies Protect Mice against *Candida auris* Invasive Infection

**DOI:** 10.3390/ijms22116162

**Published:** 2021-06-07

**Authors:** Jonothan Rosario-Colon, Karen Eberle, Abby Adams, Evan Courville, Hong Xin

**Affiliations:** Department of Microbiology, Immunology, and Parasitology, Louisiana State University—Health Sciences Center, New Orleans, LA 70112, USA; jcolo1@lsuhsc.edu (J.R.-C.); keberl@lsuhsc.edu (K.E.); abbya@lsu.edu (A.A.); ecour2@lsuhsc.edu (E.C.)

**Keywords:** *C. auris*, candidiasis, multidrug resistance, monoclonal antibodies, universal antibodies, cell wall, passive immunization

## Abstract

*Candida auris* is a multidrug-resistant fungal pathogen that can cause disseminated bloodstream infections with up to 60% mortality in susceptible populations. Of the three major classes of antifungal drugs, most *C. auris* isolates show high resistance to azoles and polyenes, with some clinical isolates showing resistance to all three drug classes. We reported in this study a novel approach to treating *C. auris* disseminated infections through passive transfer of monoclonal antibodies (mAbs) targeting cell surface antigens with high homology in medically important *Candida* species. Using an established A/J mouse model of disseminated infection that mimics human candidiasis, we showed that C3.1, a mAb that targets β-1,2-mannotriose (β-Man_3_), significantly extended survival and reduced fungal burdens in target organs, compared to control mice. We also demonstrated that two peptide-specific mAbs, 6H1 and 9F2, which target hyphal wall protein 1 (Hwp1) and phosphoglycerate kinase 1 (Pgk1), respectively, also provided significantly enhanced survival and reduction of fungal burdens. Finally, we showed that passive transfer of a 6H1+9F2 cocktail induced significantly enhanced protection, compared to treatment with either mAb individually. Our data demonstrate the utility of β-Man_3_- and peptide-specific mAbs as an effective alternative to antifungals against medically important *Candida* species including multidrug-resistant *C. auris*.

## 1. Introduction

*Candida auris* is an emerging fungal pathogen first identified in Tokyo, Japan in 2009 [1]. It has since emerged throughout much of the world, with many countries reporting multiple clinical cases [2]. Unlike other pathogenic *Candida* species, *C. auris* has a propensity to colonize abiotic surfaces as well as the human skin [3]. This makes the nosocomial spread of the pathogen especially prevalent and contributes to a higher potential to disseminate into bloodstream infections compared to other *Candida* species. Consequently, ICU patients and nursing home residents are highly vulnerable to nosocomial infections with *C. auris*, and migration from the skin to a disseminated bloodstream infection is especially common in patients with underlying comorbidities, those under immunosuppressed conditions, or those who have undergone invasive surgical interventions [4,5]. This ease of spread has contributed to a slew of healthcare-associated outbreaks, with contamination of ICUs persisting for several weeks [2,6]. Furthermore, with the ongoing COVID-19 pandemic caused by the novel coronavirus, SARS-CoV-2, the rate of hospitalizations is currently extremely high. With many ICU units being filled to capacity, this creates the perfect environment for further *C. auris* ICU outbreaks [7,8,9]. Once systemic, *C. auris* infection is often fatal, having a case mortality rate of 33–60% [4,10,11,12], which is much higher than that of other pathogenic *Candida* species. Mortality is most often attributed to multiorgan failure, with the kidney and heart being most susceptible [13,14,15].

A defining feature of *C. auris*, among other *Candida* species, is its multidrug resistance. Although antifungal resistance has been reported in other *Candida* species, most notably with *Candida glabrata*, the degree of antifungal resistance observed in *C. auris* is unprecedented [2]. In a study that looked at antifungal resistance in 99 clinical isolates of *C. auris* from the United States, 89% of isolates were resistant to fluconazole, 30% were resistant to amphotericin B, and 6% were resistant to echinocandin drugs [16]. In another study of 1385 United States clinical isolates of *C. glabrata*, 9.6% of isolates were resistant to fluconazole and 6% were resistant to echinocandin drugs [17]. Similarly, *C. auris* clinical isolates from across the globe have consistently shown high resistance to antifungals within the azole and polyene drug classes [4,18]. Being so, the typical course of treatment for *C. auris* bloodstream infections is the daily administration of echinocandin drugs, such as micafungin, caspofungin, or anidulafungin. A major limitation of antifungals, however, is their associated drug toxicities. Immunocompromised patients, who are most susceptible to disseminated infection, are in a fragile state and often unable to tolerate additional organ toxicity caused by commonly prescribed antifungal drugs, therefore rending these drugs ineffective [19]. Furthermore, there have been several reports of *C. auris* isolates that are pan-resistant to all three major antifungal drug classes, which greatly limits treatment options [4]. Due to its high degree of antifungal resistance, the potential to spread throughout the hospital environment, and its associated high mortality rate, *C. auris* is the first fungal pathogen to be labeled a serious global public health threat, and new treatments are urgently needed [20].

To overcome the problems of *C. auris* antifungal resistance and drug toxicity, we sought to investigate if prophylactic treatment using *Candida*-specific monoclonal antibodies (mAbs) could induce protection against *C. auris* bloodstream infections in A/J mice, as an alternative to conventional antifungal drug treatment. Protective mAb therapy is an emerging, yet highly promising strategy for the treatment of microbial diseases [21,22]. Antibodies are known to confer protection to various pathogens via several mechanisms, including neutralization, opsonization, and complement activation [23]. As of today, the United States Food and Drug Administration (FDA) has approved five different synthetic mAb for the treatment of various viral and bacterial diseases, including human immunodeficiency virus (HIV) and *Clostridioides difficile* infections [24]. Additionally, two new synthetic mAb-based treatments, Eli Lilly’s Bamlanivimab and Regeneron’s REGN-COV2 cocktail, are currently in clinical trials and have shown promising efficacy against COVID-19 [25], and the FDA has approved both drugs for emergency use authorization (EUA) [26,27]. Presently, there are no mAb-derived drugs for the treatment of fungal diseases, even within clinical trials. Particularly with pathogens such as *C. auris*, which have developed high levels of drug resistance and cause high mortality in immunocompromised patients, mAb therapy is an attractive treatment option.

Since the cell wall is the first point of contact between *Candida* and the host’s immune system, we developed “universal mAbs” that target various *Candida* cell wall epitopes that share high homology among various *Candida* species. A major benefit of universal mAbs is that they could potentially be applied for the treatment of candidemia caused by multiple pathogenic species of *Candida*, such as *Candida albicans*, *Candida glabrata*, *Candida tropicalis*, *Candida krusei*, and *C. auris*. This is especially important because infected individuals often do not receive a timely diagnosis due to unspecific symptoms of invasive candidiasis.

Overall, we hypothesized that prophylactic treatment with universal mAbs would induce extended survival and enhanced fungal clearance within an A/J mouse model of *C. auris* disseminated infection. A/J mice are deficient in complement protein C5 and its cleaved product C5a, a pro-inflammatory chemoattractant important for anti-*Candida* protection [28,29,30]. This renders A/J mice highly susceptible to *C. auris* disseminated infection without the need for immunosuppressive drugs [31]. Using this model, we identified three mAbs that provided significant protection, as evidenced by extended survival and lower fungal burdens in the kidney, brain, and heart, compared to control mice. In addition, our results showed that two of our mAbs could be administered as a cocktail to further enhance their effectiveness. Overall, our results demonstrate the efficacy of passive transfer with universal mAbs as a novel treatment against multidrug-resistant *C. auris*.

## 2. Results

### 2.1. In Vitro and In Vivo Efficacy of Antifungals against Multidrug-Resistant C. auris

*C. auris* isolates can be grouped into five clades (I-V) originating from different geographic regions [4]. Within each clade, isolates may have differences in morphology, levels of virulence, growth rates, and antifungal-resistance profiles [4,32]. Being so, in preparation for our animal studies, we first investigated the antifungal susceptibility of two clinical isolates of *C. auris* belonging to distinct clades: AR-0386 (CAU-06) of Clade IV and AR-0389 (CAU-09) of Clade I. AR-0386 is a highly aggregative South American isolate that has been shown to be less virulent than *C. albicans* in mouse models [33], while AR-0389 is a nonaggregative South Asian isolate that is highly virulent in mouse models [31,34]. It has been reported that nonaggregating *C. auris* isolates such as AR-0389 are among the most virulent clinical isolates, with virulence comparable to that of *C. albicans* in the invertebrate *Galleria mellonella* model [35].

We first performed an in vitro minimum inhibitory concentration (MIC) assay using two commonly administered antifungal drugs, micafungin, and itraconazole. These two antifungals belong to the echinocandin and azole drug classes, respectively, and isolates AR-0386 and AR-0389 have been reported to be susceptible to both drugs in vitro [34,36]. As a comparison, we also tested the MICs for *C. albicans* reference strain SC5314. After 48-h of drug exposure, the micafungin MIC50 was determined to be 0.063 μg/mL for AR-0386, 0.125 μg/mL for AR-0389, and 0.031 μg/mL for *C. albicans* (Table 1). For itraconazole, the 48 h MIC50 was 2.0 μg/mL for AR-0386, 0.25 μg/mL for AR-0389, and 0.031 μg/mL for *C. albicans*. These results showed that AR-0386 and AR-0389 were susceptible to micafungin and itraconazole in vitro, although both isolates were much more resistant to itraconazole than was *C. albicans*.

A limitation of in vitro assays is that they reflect the limited environment within the test tube, which is considerably different from the environmental conditions encountered in vivo. Being that MIC data cannot always reliably predict in vivo drug susceptibility [37], we next tested the micafungin and itraconazole susceptibility of *C. auris* using a complement C5-deficient A/J mouse model of disseminated infection [33]. To begin, we established an appropriate sublethal challenge dose that would result in 80–100% survival within 10 days post challenge using the highly virulent AR-0389 strain (Figure 1A). Our results showed that doses below 8 × 10^7^ CFUs resulted in 100% survival by day 10. Accordingly, we decided to use a dose of 4 × 10^7^ CFUs for our in vivo antifungal susceptibility study. After the challenge, mice were treated daily with a minimum protective dose of micafungin (0.25 mg/kg/day) or itraconazole (1.67 mg/kg/day), which were determined via an antifungal pilot study (data not shown). Upon termination on Day 6, micafungin-treated mice showed no reduction in fungal burdens in the kidney, brain, or heart compared to control mice (Figure 1B). Similarly, itraconazole provided no significant reduction in organ burdens using the minimum dose.

Different inbred mouse strains have differences in MHC haplotypes, immunophenotype features, and isoforms of metabolic enzymes which can contribute to varying rates of drug metabolism [38,39]. Being so, we repeated our in vivo micafungin and itraconazole susceptibility assay using an immunosuppressed C57BL/6 mouse model of disseminated infection to compare *C. auris* susceptibility to that of A/J mice. C57BL/6 mice were immunosuppressed with cyclophosphamide prior to challenge with a sub-lethal dose of *C. auris* AR-0389. Beginning 18 h post challenge, mice were administered micafungin or itraconazole daily for 14 days and then sacrificed on day 15. As with our A/J mouse model, we saw no protective effect using itraconazole, even with a higher dose (5.0 mg/kg/day) (Figure 1C). Interestingly, with the minimum dose of micafungin, we observed a significant reduction in fungal burdens in the kidney and brain (2.9 × 10^4^ and 2.2 × 10^3^ CFUs/g, respectively) by day 15 as compared to DPBS mice kidney and brain burdens (5.5 × 10^6^ and 3.4 × 10^4^ CFUs/g, respectively). Comparing these results to our MIC data showed that although *C. auris* AR-0389 is susceptible to both micafungin and itraconazole in vitro, only low-dose micafungin was effective at significantly reducing fungal burdens in immunosuppressed C57BL/6 mice. The data showed that *C. auris* is susceptible to minimum doses of micafungin in vivo, but this could vary with mouse strain and cannot be predicted solely using in vitro MIC data.

### 2.2. Candida Cell Surface Binding of Universal Candida-Specific Monoclonal Antibodies

Considering that antifungals have limited efficacy against multidrug-resistant *C. auris*, we next investigated the protective efficacy of a panel of universal monoclonal antibodies (mAbs) that target different *Candida* cell surface epitopes, which share high homology among various *Candida* species (Table 2). First, we validated antibody binding to cell surface epitopes by flow cytometric analysis using *C. auris* AR-0386 and AR-0389 and *C. albicans* SC5314. We focused on three mAbs that were shown to be protective in our preliminary studies: C3.1, which targets β-1,2-mannotriose (β-Man_3_, IgG3), a mannose sugar that is abundantly expressed and distributed on the outer cell wall of most *Candida* species [40,41]; 6H1, which targets hyphal wall protein 1 (Hwp1, IgG2a), a cell wall mannoprotein that is involved in adhesion, biofilm formation, and hyphal development in several *Candida* species [42]; 9F2, which targets phosphoglycerate kinase 1 (Pgk1, IgG1), a metabolic enzyme that is primarily involved in glycolysis and gluconeogenesis within the cytoplasm [43,44].

Isolates were incubated with each primary mAb, washed, and then incubated with goat anti-mouse secondary antibody conjugated to Alexa Fluor 488. Antibody binding was then detected using flow cytometry (Figure 2A–C). C3.1 (anti-β-Man_3_, IgG3) showed 56.6% binding to AR-0386, 98.4% binding to AR-0389, and 86.5% to *C. albicans*, while 6H1 (anti-Hwp1, IgG2b) showed binding of 38.1% to AR-0386, 2.6% to AR-0389, and 21.2% to *C. albicans*. Finally, binding of 9F2 (anti-Pgk 1, IgG2a) was at 33.0% for AR-0386, 3.4% for AR-0389, and 38.6% for *C. albicans*. Additionally, fluorescent microscopy imaging of cells stained with each antibody (Figure 2D–F) depicted levels of fluorescence that corresponded with our flow cytometry data. Since mannose sugars are abundantly expressed on the outer cell wall [40,41], a high level of C3.1 binding in all isolates was expected. Pgk1 and Hwp1, on the other hand, are not major components of the *Candida* cell wall and not as abundantly expressed as β-Man_3_, which was reflected in our mAb-binding data. It was, however, surprising to see modest Hwp1 binding, since *C. auris* has not been demonstrated to express Hwp1, and its genome has not been shown to contain an ortholog of the *Hwp1* gene. This pointed to the possibility that our anti-Hwp1 mAb could be cross-reactive with another *C. auris* cell wall protein, in which further investigation is required to identify the target as well as confirm its sequence. It was also surprising to see such disparate levels of 6H1 or 9F2 binding between the two *C. auris* isolates. The lower levels of 9F2 and 6H1 binding to AR-0389 could be an indication of epitope masking, which could explain the higher level of virulence of this isolate, compared to AR-0386. Overall, the data showed that the universal mAbs bind to *C. auris* in an isolate-specific manner.

### 2.3. In Vivo Protective Efficacy of Universal Candida β-1,2-Mannotriose- and Peptide-Specific Monoclonal Antibodies

We next evaluated if the prophylactic passive transfer of our universal mAbs would protect mice against *C. auris* disseminated infection. We began with C3.1 (anti-β-Man_3_, IgG3), a mAb that has previously been shown to be highly protective against *C. albicans* disseminated bloodstream infections in mice [45]. Since the *C. auris* cell wall has a high composition of β-1-2-mannans—reportedly even higher than in *C. albicans* [46], we hypothesized that mAb C3.1 would also protect against disseminated infection caused by *C. auris*. To compare the efficacy of C3.1 to that of micafungin, A/J mice were treated one time with C3.1 or DPBS or treated daily for 7 days with micafungin. 24 h after C3.1 treatment, mice were challenged with a sublethal dose of *C. auris* AR-0386. By day 35 post infection, there was a significant increase in survival of C3.1-treated mice (100% survival), compared to DPBS control mice (40% survival) (Figure 3A). Micafungin-treated mice survival (60% survival) was not significantly extended, compared to the DPBS group. When quantifying fungal burdens, C3.1-treated mice had a significant reduction in kidney and brain burdens (1.2 × 10^4^ and 5.0 × 10^1^ CFUs/g, respectively), compared to control mice (6.6 × 10^8^ and 6.2 × 10^6^ CFUs/g, respectively) (Figure 3B). Remarkably, all C3.1-treated mice had undetectable brain burdens by day 35. While there was also a reduction in heart burdens in C3.1-treated mice (8.2 × 10^2^ CFUs/g), compared to DPBS control (4.4 × 10^6^ CFUs/g), this change was not statistically significant. Consistent with survival data, there was no significant reduction in the kidney, brain, or heart burdens in micafungin-treated mice (2.8 × 10^8^, 4.9 × 10^4^, and 4.4 × 10^6^ CFUs/g, respectively), as compared to DPBS mice. The data showed that mAb C3.1 treatment outperformed low-dose micafungin, a gold-standard drug for the treatment of invasive *C. auris* infection in an A/J mouse model.

We then evaluated the prophylactic efficacy of our peptide-specific mAbs. In addition to the mAbs 6H1 and 9F2, we also screened one additional universal mAb, 10E7, which targets a different epitope on Pgk1 (GPV-P3, IgG1). A/J mice were treated with each mAb, followed by a lethal dose challenge with *C. auris* AR-0386 18 h later. Survival was observed for 40 days, and fungal burdens were quantified in the kidney, brain, and heart. With this lethal challenge dose, all control mice died on day 5 post challenge (Figure 4A). On the other hand, 9F2 and 10E7 mice had prolonged survival (50% and 25% survival, respectively) by day 40, with 9F2 inducing significantly higher survival, compared to control mice. Although not statistically significant, both mAb-treated groups had slightly reduced fungal burdens in the kidneys, with 9F2 also inducing significantly lower heart burdens (2.5 × 10^7^ CFUs/g), compared to DPBS mice (2.1 × 10^8^ CFUs/g) (Figure 4B). This reduction in heart burdens is host significant, because A/J mice ultimately succumb to cardiac failure, therefore making the heart the best indicator for evaluating disease progress and protection [33,47]. There was no difference in brain burdens among mAb-treated and control groups.

In a separate experiment, we also tested the prophylactic protective efficacy of 6H1 (anti-Hwp1, IgG2b) in A/J mice using a sublethal challenge dose of *C. auris* AR-0389. Mice were sacrificed on day 6, and fungal burdens were quantified as previously (Figure 4C). Based on the data, 6H1-treated mice had lower fungal burdens in the kidney, brain, and heart (3.8 × 10^7^, 6.7 × 10^5^, and 1.5 × 10^7^ CFUs/g, respectively), compared to DPBS mice kidney, brain, and heart burdens (2.9 × 10^8^, 1.8 × 10^6^, and 1.8 × 10^8^ CFUs/g, respectively). Of these, only the kidneys had a statistically significant reduction in fungal burdens, while the reduction in heart burdens was nearly significant (*p* = 0.0798). Collectively, the data showed that passive transfer of universal *Candida* peptide-specific mAbs, 9F2 and 6H1 provided significantly extended survival (9F2) and significantly reduced fungal burdens in the kidney (6H1) and heart (9F2), compared to control mice in an A/J mouse model of *C. auris* disseminated infection.

### 2.4. In Vivo Protective Efficacy of Monoclonal Antibody Cocktails

Being that several of our antibodies were able to induce protection in mice when administered individually, we further tested if combining two different mAbs would induce enhanced or even synergistic protection in mice. Since our mAbs are specific to different cell surface antigens, we hypothesized that using a cocktail of two mAbs could result in a more effective “double hit” protection against *C. auris*. The selected two-mAb cocktail consisted of 6H1, which performed best in the kidney (Figure 4C), and 9F2, which performed best in the heart (Figure 4B). A/J mice were treated with 6H1, 9F2, or both, followed by a sublethal dose challenge of AR-0386 24 h later. Survival was observed for 35 days. Mice that received the 6H1+9F2 cocktail had significantly higher survival (100% survival) by day 35, compared to mice that received only 6H1 (20% survival) or 9F2 (0% survival) (Figure 5A). Regarding fungal burdens (Figure 5B), mice that received the cocktail had significantly lower burdens in the kidney (1.5 × 10^8^ CFUs/g), compared to mice that received only 9F2 (1.0 × 10^9^ CFUs/g). Kidney burdens in the cocktail group were also lower than in the 6H1 group (6.7 × 10^8^ CFUs/g), although this was not statistically significant. Similarly, fungal burdens in the brain were significantly lower in mice that received the cocktail (5.8 × 10^3^ CFUs/g), compared to mice that received only 9F2 (9.3 × 10^6^ CFUs/g). The cocktail group brain burdens were also lower than the 6H1 group (3.4 × 10^4^ CFUs/g), although this was not statistically significant. We also observed a consistent trend of lower heart burdens for the mAb cocktail treated group (1.1 × 10^4^ CFUs/g), compared to 6H1 mice (3.3 × 10^6^ CFUs/g) or 9F2 (8.3 × 10^6^ CFUs/g), although not statistically significant. Overall, the data showed that protective mAbs, such as 6H1 and 9F2, can be combined into cocktails to provide enhanced protection against *C. auris* disseminated infection compared to treatment with either mAb individually.

## 3. Discussion

*C. auris* is the first fungal pathogen to cause a serious global public health threat [20]. In vitro and in vivo studies of drug efficacy against this MDR pathogen have shown that most isolates are highly resistant to azoles and polyenes, which severely limits effective drug treatments. As this pathogen easily spreads throughout hospital ICUs and assisted living facilities, *C. auris* seriously threatens the lives of patients living with comorbidities. In those who are most at risk, such as the immunocompromised, antifungal drugs are often not effective and cause additional organ toxicity, which can further exacerbate the conditions of these highly susceptible patients [19].

In this study, we demonstrated for the first time that passive transfer of *Candida* peptide- and glycan-specific universal mAbs is an effective means of immunotherapy for protecting against *C. auris* invasive infections in an established A/J mouse model. During systemic infection, *Candida spp.* disseminates through the bloodstream and enters target organs within hours of infection [48]. During early infection, *Candida* cells are rapidly eliminated from circulation, and the pathogen is often undetectable in the blood [49]. Consequently, our study evaluated antibody efficacy by quantifying fungal burdens in the kidney, brain, and heart, as well as overall survival. A key characteristic of our mAbs is that they target epitopes that share high homology among various clinically significant *Candida* species. This “universal” targeting allows for their potential application in preventing candidemia caused by different *Candida* species and isolates regardless of isolate-specific antifungal-resistance profiles.

The two *C. auris* isolates analyzed, AR-0386 (CAU-06) and AR-0389 (CAU-09), come from different clades and have unique genetic variations resulting in isolate-specific morphologies, rates of proliferation, virulence, and antifungal resistance [32]. These genetic differences may also affect cell wall composition. The lower level of antibody binding observed with isolate AR-0389 could be due to epitope masking by cell wall polysaccharides, which could account for evasion of immune responses and higher levels of virulence observed with isolate AR-0389, compared to AR-0386. This hypothesis is supported by evidence showing that the unmasking of cell wall components, such as β-(1,3)-glucan, can lead to attenuated *C. albicans* virulence in mouse models of systemic infection [50,51]. More work is required to determine why the same mAb has different binding patterns to the cell surface epitopes of *C. auris* isolates, and how this is related to isolate virulence, host–pathogen interaction, and immune escape.

Of our mAb panel, C3.1 (anti-β-Man_3_) had the highest level of cell surface binding, provided the best overall protection, and significantly outperformed micafungin as a treatment for invasive *C. auris* infection. β-mannose is a major glycan component of the outer cell wall of many *Candida spp*., and it is abundantly expressed on the surface of *C. auris* [46]. Our previous data with *C. albicans* has shown that C3.1-mediated protection depends on its ability to rapidly and efficiently fix complement to the fungal surface, which is associated with enhanced phagocytosis and killing of the fungus [40,45]. Being that A/J mice are C3-competent, C3.1 likely protects against *C. auris* via the same mechanism. Furthermore, it has been shown that immune complexes of IgG3 can bind to FcγRI receptors on phagocytic cells [52]. Thus, C3.1 opsonization of *C. auris* may lead to additional FcγRI binding, increased adherence and internalization, and enhanced phagocytosis; however, quantitation of these values will be the subject of a later study.

In contrast to C3.1, 6H1 (anti-Hwp1) and 9F2 (anti-Pgk1) mAbs target epitopes that are not abundantly expressed in the *Candida* cell wall, which was reflected in our flow cytometry data. Although Pgk1 is primarily a glycolytic enzyme localized in the cytoplasm, it is also exposed on the *Candida* surface and has been identified as a cell-wall-associated moonlight protein that is immunoreactive during invasive fungal infections in humans [43,44,53]. Hwp1, on the other hand, is expressed on the cell surface during hyphal morphology [54]; however, some evidence does suggest that it may also be expressed during pseudohyphal morphology [55], which has been induced in *C. auris* in vitro [56]. Interestingly, the *C. auris* genome does not appear to contain an ortholog of the Hwp1 gene [54,56], which points to the possibility of cross reaction of the anti-Hwp1 mAb with another cell wall protein. Nonetheless, both mAbs were able to induce significant protection in our mouse model.

Our animal model also showed the enhanced efficacy of two-mAb cocktails as a prophylactic treatment against *C. auris* disseminated infection. A benefit of mAb cocktails is that each antibody can individually induce a different effector mechanism, which can work in concert to inhibit the growth and dissemination of the pathogen [57]. This is a strategy that has been highly effective in treating other infectious diseases, such as HIV infection using highly active antiretroviral therapy (HAART), which uses a cocktail of three or more drugs that inhibit different steps of the virus’s replication cycle [58]. In the case of the 6H1+9F2 cocktail, the enhanced protection could be due to improved access of phagocytic cells to *C. auris* yeast, leading to increased oxidative damage. Research conducted by other groups has shown that Pgk1 confers protection to *C. albicans* against reactive oxygen species (ROS) [59] and that immunization with recombinant Pgk1 protein leads to a significant reduction in kidney burdens in mice infected with *C. albicans* or with *C. glabrata* [60]. Additionally, in an in vivo rat venous catheter model of infection, *C. albicans* Hwp1 mutants were shown to display severe biofilm defects with few hyphae [61]. This may indicate that 6H1 and 9F2 mAbs could function together to disrupt biofilm formation and increase susceptibility to respiratory burst by phagocytic cells, although this mechanism would need to be further investigated. Alternatively, the binding of one mAb could induce a conformational change that results in the unmasking of the second mAb’s target epitope, leading to functional cooperativity between mAbs targeting different epitopes. Further investigation may also show that these different mechanisms are not mutually exclusive and may both contribute to the observed enhanced efficacy.

In recent years, several novel antifungal compounds have been developed that have proven effective against MDR *Candida* species. One such compound, carvacrol, was shown to have antifungal activity against clinical isolates of *C. auris* while also inducing synergistic antifungal activity when combined with fluconazole, amphotericin B, caspofungin, and micafungin [62]. As with these compounds, mAbs have the potential to be combined with antifungals to induce synergistic protection while also significantly reducing drug MICs and associated toxicity. Future experiments will evaluate the enhanced protection of combining our mAbs with conventional antifungals as well as the therapeutic efficacy of these mAbs.

Finally, it is important to note that similar to most pathogenic *Candida* species, *C. auris* has the propensity to form biofilms within its host. It is well established that within biofilms, *Candida spp.* exhibit higher resistance to antifungal drugs. This is largely due to the upregulation of efflux pumps [63] and the production of extracellular polymeric substances (EPS), which can interfere with drug diffusion [64]. Although we observed that micafungin was effective at reducing fungal burdens within our immunosuppressed C57BL/6 model, this efficacy would likely be reduced against biofilm-derived *C. auris* cells. In one mouse study of disseminated candidiasis using biofilm-derived *C. glabrata* cells, micafungin treatment was ineffective at reducing liver and kidney burdens [65]. MAbs could be effective here as well. In one study, after incubating *C. albicans* and *Candida dubliniensis* with antibodies targeting the surface antigen, complement receptor 3-related protein (CR3-RP), both species had a reduction in surface adherence and in biofilm thickness in vitro [66]. Using a combination therapy, the binding of mAbs to the surface of the pathogen could interfere with biofilm formation, leading to increased diffusion of antifungals.

In summary, the potential for mAb therapy against microbial pathogens is vast since mAbs inherently have high specificity for their targets without selecting for resistance. The application of protective mAbs against *C. auris* disseminated infection represents a highly promising alternative to the often-ineffective use of antifungal drugs against this MDR pathogen. Not only can effective mAbs protect against severe infection more rapidly than antifungal drugs [67,68,69], but specific antibodies may also be synergistic with conventional antimicrobials. The data presented here have significant implications for both immunotherapy and vaccine development in the future, and the demonstration of preclinical efficacy of the immunoprotective mAbs in this study will provide compelling data that can be advanced into the clinical setting.

## 4. Materials and Methods

### 4.1. Candida Isolates and Culture Conditions

Two antifungal-resistant isolates of *C. auris*, AR-0386 (CAU-06) and AR-0389 (CAU-09), were supplied by the United States Centers for Disease Control and Prevention (CDC, Atlanta, GA, USA). *C. albicans* reference strain SC5314 was supplied by the American Type Culture Collection (MYA-2876, ATCC, Manassas, VA, USA). For passive transfer of mAb experiments, the inoculum was serially passaged daily for three days in 25 mL glucose yeast peptide broth at 37 °C and then washed three times in Dulbecco’s phosphate-buffered saline (DPBS). Cell density was measured using a hemocytometer and adjusted to the desired density in DPBS. For MIC assays, culture was plated onto Sabouraud Dextrose Broth (SDB) agar plates and incubated at 35 °C for 24 h. Five colonies were selected and suspended in 1 mL of sterile deionized H_2_O, and cell density was measured using a hemocytometer. The density was then adjusted to desired concentration in RPMI 1640 + 0.165 M MOPS medium (with L-glutamine and phenol red, without bicarbonate).

### 4.2. Mice

Male and female A/J mice were purchased from the Jackson Laboratory (JAX, Bar Harbor, ME, USA) and Envigo (Indianapolis, IN, USA). Male and female C57BL/6 mice were purchased from the Jackson Laboratory (JAX, Bar Harbor, ME, USA). At the time of studies, A/J mice were 7 weeks old, and C57BL/6 mice were 16–17 weeks old. Mice were maintained in the Louisiana State University Health Sciences Center’s AAALAC-accredited animal facility (#000037, LSUHSC-NO, New Orleans, LA, USA), and all animal experiments were performed using a protocol approved by the Louisiana State University Health Sciences Center’s Institutional Animal Care and Use Committee (#3559, 1/18/2019, LSUHSC-NO IACUC, New Orleans, LA, USA).

### 4.3. Immunosuppression

16–17-week-old male and female C57BL/6 mice were immunosuppressed using cyclophosphamide monohydrate (#C0768, Sigma-Aldrich, St. Louis, MO, USA) three days prior to challenge by intraperitoneal (i.p.) injection using a dose of 200 mg/kg of body weight. Immunosuppression was maintained with additional i.p. injections of a 150 mg/kg dose of cyclophosphamide every 7 days.

### 4.4. Antifungals

Micafungin (≥97% HPLC) was purchased from Sigma-Aldrich (#SML2268, Sigma-Aldrich, St. Louis, MO, USA), and itraconazole (≥98% TLC) was purchased from Sigma-Aldrich (#I6657, Sigma-Aldrich, St. Louis, MO, USA). For MIC assay, micafungin and itraconazole were dissolved in RPMI 1640 + 0.165 M MOPS medium (with L-glutamine and phenol red, without bicarbonate) + 1% DMSO to the desired concentrations. For animal experiments, micafungin was dissolved in DPBS to the desired concentration, and itraconazole was dissolved in sterile deionized H_2_O + 10% DMSO to the desired concentration.

### 4.5. Antifungal Susceptibility

*C. auris* and *C. albicans* micafungin and itraconazole minimum inhibitory concentrations (MICs) were determined using the broth microdilution method (BMD) according to the guidelines of the Clinical and Laboratory Standards Institute (CLSI) Reference Method for Broth Dilution Antifungal Susceptibility Testing of Yeasts; Approved Standard (CLSI M27-A3, 3rd Ed. 2008. Vol 28, No 14).

### 4.6. Antibodies

All monoclonal antibodies were isolated from hybridoma cells, purified, and sterile filtered in phosphate-buffered saline (PBS) by GenScript Biotech Corporation (GenScript, Piscataway, NJ, USA) and by Autoimmune Technologies (AiT, New Orleans, LA, USA). Antibody concentrations were determined using a Pierce BCA Protein Assay Kit (#23227, Thermo Scientific, Waltham, MA, USA), according to the manufacturer’s directions. A standard curve was constructed using bovine gamma globulin standard (BGG) (#23212, Thermo Scientific, Waltham, MA, USA). Absorbance was read on a plate reader at 562 nm, and sample absorbances were compared to the BGG standard curve to determine antibody concentration.

### 4.7. Antibody Titers

Titers of mAbs were determined via enzyme-linked immunosorbent assay (ELISA). A 96-well polystyrene plate was coated with whole synthetic protein (Pgk1, Hwp1) (GenScript, Piscataway, NJ, USA) or mannan extract at 4 μg/mL in bicarbonate coating buffer and incubated overnight at 4 °C. The following day, the wells were blocked using 1% BSA blocking buffer for 1 h at room temperature. Monoclonal antibodies were then added in duplicate to respective wells using twofold serial dilutions from 1:500 to 1:256,000 and incubated for 2 h at room temperature. Horseradish peroxidase (HRP)-conjugated goat anti-mouse polyvalent IgG, IgA, IgM secondary antibody (#A0412, Sigma-Aldrich, St. Louis, MO, USA) was then added (1:3000) and incubated in the dark for 1 h at room temperature. Tetramethyl benzidine (TMB) (#34022, Thermo Scientific, Waltham, MA, USA) was then added to each well and incubated in the dark for 30 min at room temperature. HCl was added to stop the reaction, and the optical density was measured at 450 nm using a spectrophotometer. As a negative control, wells were incubated with secondary antibody alone. Titers were given as the dilution whose OD reading was greater than two times that of the negative control.

### 4.8. Antibody Cell Surface Staining

Overnight cultures of *C. auris* AR-0386, AR-0389, and *C. albicans* SC5314 were washed three times with DPBS. Pellets were resuspended in 100 μL of C3.1 (anti-β-Man_3_), 9F2 (anti-Pgk1), or 6H1 (anti-Hwp1) antibodies in 1X PBS + 1% BSA and incubated for 1 h at room temperature. Cells were washed three additional times, and pellets were resuspended in Alexa Fluor 488-conjugated goat anti-mouse IgG, IgM secondary antibody (#A10680, Invitrogen, Carlsbad, CA, USA) (1:100) in 1X PBS + 1% BSA and incubated for 1 h at room temperature. Cells were washed three additional times and resuspended in 500 μL DPBS and acquired by flow cytometry at 488 nm (FACSDiva 8.0.3, FACSCanto II, BD Biosciences, San Jose, CA, USA), As a positive control, an additional high-binding anti-β-Man_3_ mAb, G11.2 (IgG1), was used. As a negative control, cells were stained with secondary antibody alone. Gating was set on the secondary antibody (Alexa Fluor 488 only) control. A portion of the stained cells was spread on slides for fluorescent imaging.

### 4.9. In Vivo Model of Disseminated Infection

For this, 7-week-old A/J mice or 16–17-week-old immunosuppressed C57BL/6 mice were treated via intraperitoneal (i.p.) injection with 200 μL of monoclonal antibody or DPBS. Then, 18 h later, mice were challenged via intravenous (i.v.) injection in the tail vein with *C. auris* AR-0386 at a sublethal dose of 4 × 10^7^ CFUs in 100 μL DPBS or a lethal dose of 1 × 10^8^ CFUs in 100 μL DPBS or with *C. auris* AR-0389 at a sublethal dose of 4 × 10^7^ in 100 μL DPBS, depending on the experiment. For experiments that measured antifungal efficacy, mice received daily i.p. administration of micafungin or itraconazole starting 24 h post challenge. All mice were monitored daily for death or the development of a moribund state, at which point they were sacrificed via CO_2_ inhalation. All surviving mice were sacrificed at the conclusion of each study.

### 4.10. Quantification of Fungal Burdens

Upon death, the kidney, brain, and heart were extracted from mice, and each organ was homogenized in DPBS. The homogenate was then serial diluted and plated onto GYEP agar plates containing chloramphenicol. The plates were incubated for 48 h at 37 °C at which time CFUs were quantified. The limit of detection was 50 CFUs/g for each organ.

### 4.11. Statistical Analysis

Plots and statistical comparisons were performed using Prism Software (Version 9, GraphPad Software, San Diego, CA, USA). Survival data was evaluated by Kaplan–Meier analysis, and statistical significance was calculated using a log-rank (Mantel–Cox) test. For fungal burden data, results were expressed as mean ± SD, and statistical significance was calculated using a two-tailed *t*-test to compare mAb-treated groups to the control group. Each study contained five mice per group unless otherwise stated. Significant *p* values were defined as follows: * *p* < 0.05; ** *p* < 0.01.

## Figures and Tables

**Figure 1 ijms-22-06162-f001:**
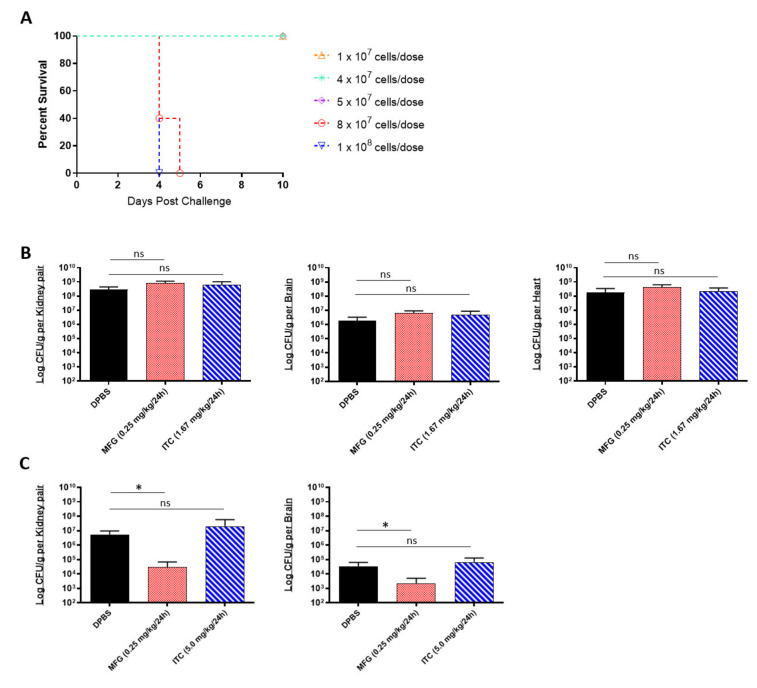
In vivo efficacy of antifungals against multidrug-resistant *C. auris*: (**A**) the 10-day survival curve of 7-week-old female A/J mice challenged with *C. auris* AR-0389 doses ranging from 1 × 10^7^ to 1 × 10^8^ CFUs; (**B**) quantification of kidney, brain, and heart fungal burdens from 7-week-old female A/J treated daily for 5 days with a minimum protective dose of micafungin or itraconazole. Mice were challenged with a sub-lethal dose of 4 × 10^7^ CFUs of *C. auris* isolate AR-0389. Starting 24 h later, mice received daily administration of 200 μL of DPBS, micafungin (0.25 mg/kg body weight), or itraconazole (1.67 mg/kg body weight). Mice were sacrificed on day 6 post challenge; (**C**) quantification of kidney and brain fungal burdens from 16- to 17-week-old male and female neutropenic C57BL/6 mice treated daily for 14 days with a minimum protective dose of micafungin or itraconazole. To induce neutropenia, mice were administered cyclophosphamide (200 mg/kg body weight) on day 3 and every 7 days after (150-mg/kg body weight). On day 0, mice were challenged with a sublethal dose of 4 × 10^7^ CFUs of *C. auris* isolate AR-0389. Starting 24 h later, mice received daily administration of 200 μL of DPBS, micafungin (0.25 mg/kg body weight), or itraconazole (5.0 mg/kg body weight). Mice were sacrificed on day 15 post challenge. MFG = micafungin, ITC = itraconazole. Data are mean + SD (**B**,**C**). *n* = 4 (**B**) *n* = 5 (**A**,**C**). Log-rank (Mantel–Cox) test (**A**) or two-tailed *t*-test (**B**,**C**) were used to identify significant differences. * *p* < 0.05; *ns* = not significant.

**Figure 2 ijms-22-06162-f002:**
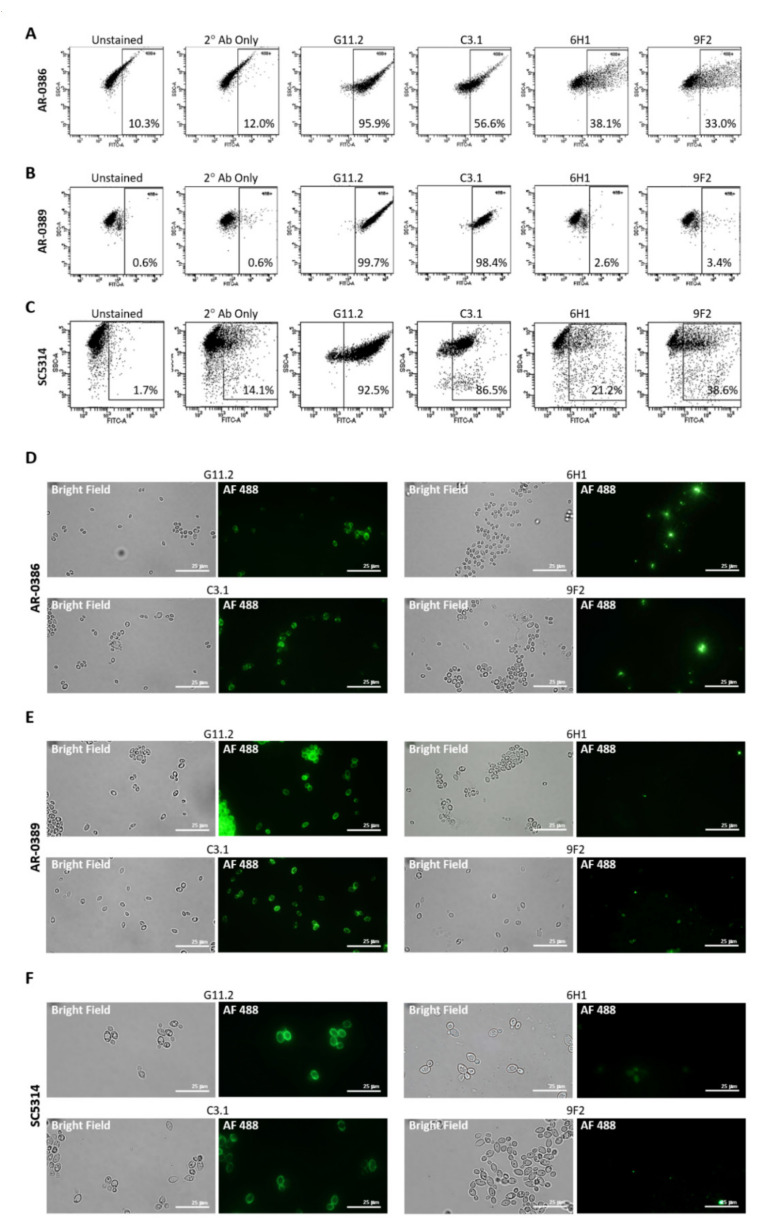
Cell surface binding of universal *Candida*-specific monoclonal antibodies: (**A**–**C**) flow cytometry scatter plots depicting levels of cell surface binding of monoclonal antibodies as a measure of Alexa Fluor 488 expression in (**A**) *C. auris* isolate AR-0386, (**B**) *C. auris* isolate AR-0389, and (**C**) *C. albicans* isolate SC5314; (**D**–**F**) confocal microscopy analysis (1000X) showing antibody cell surface staining of (**D**) *C. auris* isolate AR-0386, (**E**) *C. auris* isolate AR-0389, and (**F**) *C. albicans* isolate SC5314 using mAbs C3.1, 6H1, and 9F2. G11.2 = β-1,2-mannotriose (IgG1), C3.1 = β-1,2-mannotriose (IgG3), Hwp1 = hyphal wall protein 1, Pgk1 = phosphoglycerate kinase 1. Bar = 25 μm.

**Figure 3 ijms-22-06162-f003:**
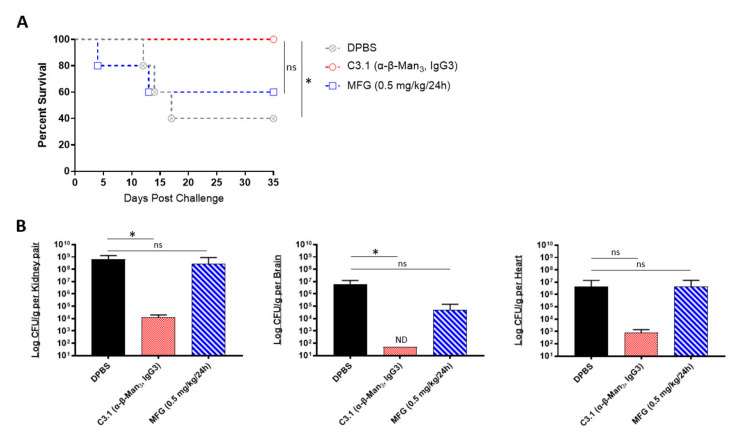
In vivo protective efficacy of universal *Candida* β-1,2-mannotriose-specific monoclonal antibody: (**A**) the 35-day survival curve and (**B**) quantification of kidney, brain, and heart fungal burdens from 7-week-old female A/J mice treated with mAb C3.1 or micafungin. Mice were treated with 200 μL DPBS, mAb C3.1 (0.24 mg/200 μL DPBS), or micafungin (0.5 mg/kg body weight daily for 7 days) then challenged 18 h later with a sub-lethal dose of 4 × 10^7^ CFUs of *C. auris* AR-0386. β-Man_3_ = β-1,2-mannotriose. Data are mean + SD (**B**). *n* = 5. Log-rank (Mantel–Cox) test (**A**) or two-tailed *t*-test (**B**) were used to identify significant differences. * *p* < 0.05; *ns* = not significant. ND = not detectable.

**Figure 4 ijms-22-06162-f004:**
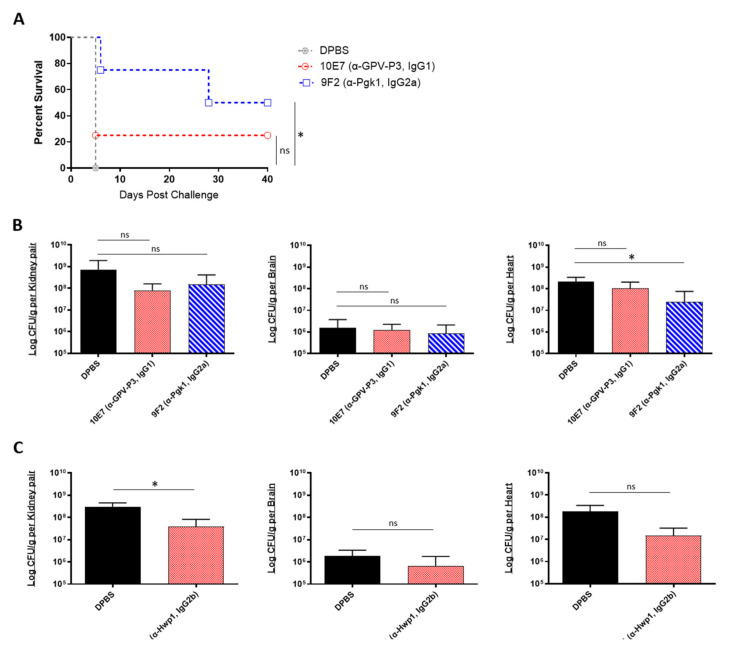
In vivo protective efficacy of universal *Candida* peptide-specific monoclonal antibodies: (**A**) the 40-day survival curve and (**B**) quantification of kidney, brain, and heart fungal burdens from 7-week-old male A/J mice treated with two Pgk1-specific mAbs; (**A**,**B**) mice were treated with 200 μL DPBS or antibody (0.285 mg/200 μL DPBS), then challenged 18 h later with a lethal dose of 1 × 10^8^ CFUs of *C. auris* AR-0386; (**C**) quantification of kidney, brain, and heart fungal burdens from 7-week-old female A/J mice treated with mAb 6H1 or micafungin. Mice were treated with 100 μL DPBS or mAb (0.135 mg/100 μL DPBS), then challenged 18 h later with a sublethal dose of 4 × 10^7^ CFUs of *C. auris* AR-0389. GPV-P3 = phosphoglycerate kinase 1 (IgG1), Pgk1 = phosphoglycerate kinase 1 (IgG2a). Data are mean + SD (**B**,**C**). *n* = 4. Log-rank (Mantel–Cox) test (**A**) or two-tailed *t*-test (**B**,**C**) were used to identify significant differences. * *p* < 0.05; *ns* = not significant.

**Figure 5 ijms-22-06162-f005:**
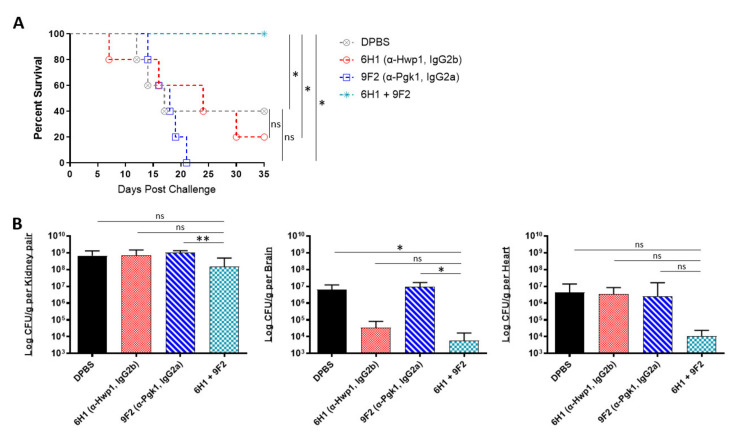
In vivo protective efficacy of monoclonal antibody cocktails: (**A**) the 35-day survival curve and (**B**) quantification of kidney, brain, and heart fungal burdens from 7-week-old female A/J mice treated with mAbs 6H1, 9F2, or a cocktail of 6H1 + 9F2. Mice were treated with two 400-μL-doses of mAb 6H1 given 18 h apart, one 200-μL-dose of mAb 9F2, or a combination of both mAbs consisting of one dose of 9F2 and two doses of 6H1 given 18 h apart. Then, 18 h after first dose of mAb, mice were challenged with a sublethal dose of 4 × 10^7^ CFUs of *C. auris* AR-0386. Hwp1 = hyphal wall protein 1, Pgk1 = phosphoglycerate kinase 1. Data are mean + SD (B). *n* = 5. Log-rank (Mantel–Cox) test (**A**) or two-tailed *t*-test (**B**) were used to identify significant differences. * *p* < 0.05; ** *p* < 0.01; *ns* = not significant.

**Table 1 ijms-22-06162-t001:** Micafungin and itraconazole MIC50 for *C. auris* isolates AR-0386 and AR-0389 and *C. albicans* isolate SC5314 at 24 and 48 h.

Drug	AR-0386 MIC50 (μg/mL)		AR-0389 MIC50 (μg/mL)		SC5314 MIC50 (μg/mL)	
	24 h	48 h	24 h	48 h	24 h	48 h
Micafungin	0.031	0.063	0.063	0.125	0.031	0.031
Itraconazole	2.0	2.0	2.0	0.25	0.031	0.031

**Table 2 ijms-22-06162-t002:** Universal *Candida* monoclonal antibodies and their cell surface targets.

Universal Antibody	Isotype	Cell Surface Target
C3.1 (anti-β-Man_3_)	IgG3	β-1,2-mannotriose
6H1 (anti-Hwp1)	IgG2b	Hyphal wall protein 1
9F2 (anti-Pgk1)	IgG2a	Phosphoglycerate kinase 1
10E7 (anti-GPV-P3)	IgG1	Phosphoglycerate kinase 1 ^1^

^1^ The 9F2 and 10E7 antibodies target two different epitopes on Pgk1.

## Data Availability

Data supporting reported results are available upon request (jcolo1@lsuhsc.edu).

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
