# Peer review of "Candida* Cell-Surface-Specific Monoclonal Antibodies Protect Mice against *Candida auris* Invasive Infection"

_ijms, 2021, doi:10.3390/ijms22116162_

Round 1
Reviewer 1 Report
This work discusses the potential therapeutic approach of a mixture of monoclonal Ab to treat the recent and highly resistant fungal pathogen – C. auris. The MS indicates that β-Man3- and peptide-specific mAbs can be a future effective alternative to common antifungals against C. auris.
This work is well performed has important results, since this pathogen has been responsible for several global outbreaks with a high patient death percentage.
The major issue this reviewer finds is that the discussion sections needs to be deeper and better examine these results with other works.
Other points are:
Introduction:
- These recent works would enrich the MS: doi.org/10.3390/ijms22094470, doi: 10.1111/myc.12904; doi: 10.1371/journal.pone.0233102;
- “A unique feature of C. auris among other Candida species is its multidrug resistance.” – this sentence is discussible as it is. Candida glabrata has also very serious multi and pan-resistance issues. Please re-phrase;
- Please write the entire name of the species, before using the abbreviated form (“species of Candida, such as C. albicans, C. glabrata, C. tropicalis, C. krusei, and C. auris.”; “G. mellonella” – check the entire MS for this;
- “2.1. In vitro and in vivo efficacy of antifungals against multidrug-resistant C. auris” – when using italic form in the title, the name of the species needs to be in non-italic form. Check the entire MS;
M&M:
- Brand and country of manufacturers need to be placed in every reagent and material (some are lacking). Please check the entire MS and adjust;
- Reference of the Ethical authorization for the in vivo study needs to be indicated;
- Version of the CLSI guidelines used in the MIC determination (put reference)?
- What was the software used to calculate the power of the in vivo assay? (e.g. GPower?);
Results and Discussion:
- In MIC determination, why use itraconazole, as it is known to have a very high resistance rate among Candida species? And why not fluconazole?
- In another in vivo assay, but with C. glabrata, micafungin did not show reduction of Candida CFUs in liver or kidneys (doi: 10.3390/jcm8020142). The present work should be compared and discussed with this one;
- The gating strategy used on cytometry assays needs to be clarified (supplementary material?);
- Figure 2 – measure bars are missing. Please put them;
- “Alex Fluor” – Do you mean Alexa Fluor 488?
Reviewer 2 Report
General Impression
The authors describe a comprehensive study on the effects of monoclonal antibodies on the establishment and progression of Candida auris infections in the mouse model. The manuscript presents data on antifungal susceptibility of the examined yeast strains, binding of cell wall-specific antibodies, protective effects of antibody cocktails and organ fungal burdens in antibody-treated mice. The rationales for the study are clearly outlined, the experiments are well described, thoroughly executed and the data analysis is appropriate. The interpretation of the results is supported by the evidence: The authors demonstrate that anti cell wall antibodies are effective in reducing fungal burden and survival of infected mice and provide solid evidence of antibody binding to cell wall epitopes.
Suggestions
- In this study, antibodies are given before the fungal inoculum. This somewhat limits the study’s value for the development of mAB therapies where antibodies are administered after an infection has been diagnosed. Have the authors considered administering antibodies after the infection to simulate a more realistic scenario? A clarification of this point would be beneficial.
- In several places, the authors discuss the AB’s potential in the treatment of candidemia, that is: presence of Candida in the bloodstream. It is not clear why no effort was reported to actually show the extent of candidemia of infected mice – instead the analysis is limited to the fungal burdens of kidney, brain and heart. The manuscript would improve by including an explanation of why blood samples were not included. This point should be addressed in the discussion – without an examination of blood samples, the study is limited in its potential to show “potential application in preventing candidemia” (line 340).
Round 2
Reviewer 1 Report
Dear authors,
Thank you for the adjustments and corrections.
Good luck.